# Factors Influencing Consumer Behavior and Purchasing Decisions Regarding Mud Crabs (*Scylla paramamosain*) in the Major Cities of Vietnam

**DOI:** 10.3390/foods14132198

**Published:** 2025-06-23

**Authors:** Le Ngoc Danh, Duong The Duy, Doan Hoai Nhan, Chau Thi Da

**Affiliations:** 1Can Tho University of Technology, 256 Nguyen Van Cu, Ninh Kieu District, Can Tho City 94000, Vietnam; tslengocdanh@gmail.com; 2Faculty of Economics and Finance, Ho Chi Minh City University of Foreign Languages and Information Technology, Ho Chi Minh City 72550, Vietnam; duydt@huflit.edu.vn; 3FPT University, Can Tho Campus, 600 Nguyen Van Cu Street, An Binh Wards, Ninh Kieu District, Can Tho City 901000, Vietnam; nhandh7@fe.edu.vn; 4Group of Applied Research in Advanced Materials for Sustainable Development, Faculty of Applied Sciences, Ton Duc Thang University, Ho Chi Minh City 72900, Vietnam

**Keywords:** fresh mud crab, consumer behavior, purchasing preference, factors influencing, consumer decision, largest cities in Vietnam

## Abstract

The mud crab (*Scylla paramamosain*), also known as the mangrove crab, thrives in shallow mangrove forests, subtidal zones, and muddy intertidal habitats. It is a highly valuable species in the mangroves and estuaries of tropical regions and is in high demand in Vietnam’s coastal markets. This study provides a comprehensive analysis of the key factors influencing consumer behavior and purchasing decisions regarding mud crabs in Vietnam’s three largest cities: Can Tho City, Ho Chi Minh City, and Hanoi Capital. To achieve this, the research employs rigorous analytical methods, including Cronbach’s alpha reliability test, principal component analysis, and multivariate regression analysis, to identify the primary determinants of consumer behavior and purchasing preferences for fresh mud crabs. The multiple regression analysis reveals two key factors that significantly influence consumer choices: nutritional knowledge and convenience awareness. Most of consumers perceive fresh mud crabs as superior in quality, expecting them to offer greater freshness; higher levels of protein, amino acids, and minerals; as well as excellent flesh texture and enhanced palatability. Based on these findings, two strategic directions are proposed for the sustainable development of Vietnam’s crab industry: (1) improving the nutritional quality of crab products to align with consumer expectations for health benefits and (2) enhancing the distribution network and diversifying product offerings to improve accessibility and convenience for consumers.

## 1. Introduction

The global demand for seafood products is increasing, and they play a significant role in household menus worldwide [1,2]. Currently, 62% of the seafood supply comes from wild capture and 38% from aquaculture [3]. The ability to expand seafood production depends on various factors, including natural conditions, available water surface area, and fishing and aquaculture practices [4,5]. More than 80% of global seafood products originate from Asian countries, including Vietnam, which offers favorable conditions for aquaculture development [6,7]. In 2022, Vietnam’s total aquatic product output reached 9.02 million tons, consisting of 5.2 million tons from aquaculture and 3.8 million tons from wild capture [8]. The aquaculture industry has experienced significant growth over the past five years, with *Pangasius* catfish and brackish shrimp (white leg shrimp and giant tiger prawn species) being the primary farmed species in the Mekong Delta of Vietnam [9,10,11]. However, since 2015, Vietnam has been increasingly affected by climate change, as indicated by saline intrusion, rising temperatures, irregular rainfall, and the construction of high dikes for rice cultivation protection. These environmental changes have contributed to widespread shrimp diseases [12,13]. In addition to environmental pressures, the shrimp farming sector faces challenges such as disease outbreaks and economic difficulties, underscoring its vulnerability. Consequently, The report on the needs of aquaculture farmers in the Mekong River Delta, Vietnam, in 2024, conducted by Ingrid [14], found that Vietnamese shrimp farmers have experienced a 20% to 60% decline in net earnings since the onset of the COVID-19 pandemic.

The mud crab (*Scylla paramamosain*) is an economically significant species inhabiting the mangroves and estuaries of tropical regions. It is a highly valued seafood product in Vietnam’s coastal areas and Southeast Asian markets. This species thrives under challenging conditions, demonstrating rapid growth, high tolerance to environmental fluctuations, strong disease resistance, a diverse diet, and substantial size [15,16]. Additionally, mud crabs possess reliable reproductive potential, significant economic value, and excellent post-harvest preservation qualities. Consequently, alongside shrimp, they are considered an ideal species for aquaculture [17,18,19,20]. Moreover, mud crab farming operates as a large-scale agricultural system that minimizes the use of pesticides and commercial feed. Extensive farming practices allow crab fry to develop naturally, with minimal human intervention [18,21,22]. This approach is expected to support the sustainable development of the aquaculture industry by reducing environmental harm and minimizing health risks associated with excessive antibiotic and growth stimulant use [21].

According to the statistics of the Directorate of Fisheries [23], total farmed mud crab production in the Mekong Delta, Vietnam, reached 66,000 tons in 2022, representing an approximate 19% increase compared to 2017 levels. This production is expected to continue growing through 2025. The previous studies reported that the transition from specialized shrimp farming to extensive mud crab farming initially generated substantial revenue. On average, farmers in the Mekong Delta earned aprroximately VND 30–50 million per hectare per crop [15,18,22,24,25]. However, mud crab farming in the region currently faces several challenges, e.g., there is no designated farming area plan established by local governments, breed quarantine measures are not implemented, and collaboration among stakeholders remains limited [15,26,27]. Additionally, the industry lacks a recognized brand for mud crab products, and farming practices remain unstandardized, with no established quality control measures for inputs. Most importantly, the absence of comprehensive government macro policies to regulate annual development in response to market fluctuations has made mud crab prices unpredictable, directly affecting farmers’ incomes [15,19,22,24,26]. Furthermore, mud crabs are primarily consumed domestically and are not widely exported [24,26].

Since the COVID-19 pandemic, consumers have become more health-conscious and focused on improving their quality of life, particularly through healthier diets that include nutritious foods like seafood and other aquatic animal products. This heightened health awareness has driven a shift in consumer behavior toward healthier choices. As a result, naturally farmed sea crab, which offers good health benefits, should be considered a priority in consumers’ purchasing decisions [28,29,30]. However, the quality and origin of seafood and aquatic products vary significantly, leading to inconsistencies in consumer trust [31,32,33].

Mud crabs, shrimp, and other crustaceans play a crucial role in human nutrition, and consumer purchasing decisions are influenced by factors such as brand recognition, perceived health benefits, and food safety concerns [34,35,36,37,38,39,40]. However, mud crab products are highly perishable and undergo significant quality deterioration during transportation and storage, posing additional challenges to their marketability [39,41,42]. Carlucci, Nocella [43] conducted a study reviewing seafood purchasing behavior and identified key factors influencing consumer decisions, including fish availability, price perception, and self-efficacy in preparation, convenience, eating habits, health beliefs, and sensory perception. Additional attributes affecting purchasing choices include production methods, country of origin, preservation techniques, product innovation, packaging, and eco-labeling. In recent years, consumer concerns about seafood quality have increased, particularly regarding live and frozen products. A study conducted by a data journalist in the United States examined consumer preferences for different crab sexes and found that American consumers prefer male crabs over female crabs due to their larger size and higher demand. Consequently, female crabs are sold at lower prices. Additionally, consumers tend to avoid female crabs due to the presence of roe, further influencing market trends [43,44]. In contrast, consumers in Vietnam primarily purchase fresh whole mud crabs for home preparation [4,16,24].

Over the past five years, Vietnam has introduced several important initiatives and regulations to promote the sustainable development of its aquaculture industry, ensure food safety, and strengthen urban food security. The government has issued the Aquaculture Development Strategy to 2030, with a vision to 2045 [45], aiming to increase aquaculture output to 7 million tons and exports to USD 12 billion by 2030. In addition, Vietnam is placing strong emphasis on environmental protection and disease control in aquaculture [46] (2022) With regard to food safety, Decree 124/2021/ND-CP has strengthened the penalties for food safety violations, while the Ministry of Health has updated food safety standards through Circular 31/2020/TT-BYT [47]). Amid growing urbanization, Ho Chi Minh City has launched the Urban Agriculture Development Program 2021–2030 to ensure a clean, sustainable food supply and enhance the resilience of the urban food system to challenges such as natural disasters and epidemics (Ho Chi Minh City People’s Committee, 2020). These policies aim to secure a stable, safe, and sustainable food supply for the population, particularly in large urban centers.

The objective of this study is to investigate and develop a comprehensive understanding of buyer behavior, purchasing preferences, and the key factors influencing consumer decisions regarding mud crabs (*Scylla paramamosain*) in Vietnam’s largest cities. The findings will provide valuable insights for mud crab producers, local governments, and policymakers, enabling them to better understand consumer preferences and more effectively optimize production to align with market demands for product type, quality, safety, and affordability.

## 2. Theoretical Approach and Perspective Review

Consumer behavior refers to the actions individuals take in acquiring and using goods or services, including searching, selecting, purchasing, and consuming products to meet personal needs. It not only involves the specific behaviors related to buying and using products but also the psychological and social processes that occur before, during, and after these actions [48,49,50,51,52].

The analysis of consumer behavior regarding perishable mud crab products aims to enhance profitability and reduce risks associated with unmet consumer demands, while also considering decision-making processes influenced by product attributes [42]. Research on the theory of planned behavior (TPB) explores the factors that influence consumer decision-making and purchase intentions [53]. Consumer behavior is a widely studied concept in economics, particularly in marketing. According to Blackwell, Miniard [51], “Consumer behavior encompasses all activities directly associated with the processes of searching, collecting, purchasing, owning, utilizing, and disposing of products and services”. It includes the decision-making processes that occur before, during, and after these actions. Consumer behavior examines the actions of individuals, groups, or organizations and the processes through which they select, acquire, use, and dispose of products, services, experiences, or ideas to satisfy personal and societal needs [48,54,55,56].

Many researchers use market analysis methods to study seafood consumers, aiming to identify the key factors influencing their purchasing decisions. This topic has been widely explored on a global scale [30,48,53,54,57]. Similar to studies conducted in Vietnam, scholars have also concentrated on consumer behaviors regarding safe seafood goods in other areas of the world [35,58,59,60,61]. Studies on customers’ preferences for fresh mud crabs have been undertaken in many crab-producing nations [39,40,41,62]. In Vietnam, most research on mud crabs has focused on the economic aspects of mud crab production [17,18,21,22,24].

However, no studies have yet explored the purchasing decisions of urban consumers regarding fresh mud crabs.

This research integrates the theory of reasoned action (TRA) with the theory of planned behavior (TPB) to achieve the research objectives. In the TRA model, an individual’s beliefs about a product shape their attitude toward purchasing fresh mud crab, which in turn influences their purchase intention, but not their actual purchase behavior directly. Therefore, attitudes help explain the factors affecting consumer purchase intentions, while these intentions serve as the strongest predictors of actual purchasing behavior [48]. In the TPB model, behavioral intentions are influenced by three key factors. First, attitudes refer to an individual’s favorable or unfavorable evaluation of a behavior. Second, social influence represents the perceived societal pressure to perform or avoid a specific action. Lastly, the theory of planned behavior expands on the TRA model by incorporating perceived behavioral control, which reflects the ease or difficulty of performing a behavior based on the availability of resources and opportunities [54,63].

The proposed research model identifies key factors influencing the purchase of fresh mud crab as follows: Consumer attitude factors (N1) include the “Perception of Nutritional Needs”, which encompasses health maintenance, adequate nutrition, absence of contamination, and the establishment of clear quality assessment standards [41,64,65]. Another key factor (N2), “Perception of Food Hygiene and Safety”, includes concerns about clear product origin, the absence of preservatives, the absence of parasites, and the absence of allergens [37,66,67]. The factor group (N3) “Price Awareness” includes market price comparisons, assessments of fair pricing relative to income, and evaluations of fresh product pricing [34,50,64]. Additionally, personal consumer variables such as age, gender, income, and distance influence the decision to purchase crabs [34,50,64]. The factor group (N4), related to the ease or difficulty of acquiring a product, includes the “Perception of Ease of Access”, which encompasses convenient storage, a variety of options, easy purchasing when needed, and efficient processing [34,35,66]. The proposed research models are presented in Figure 1. The control variables group includes age, gender, education level, and income, which are personal characteristics of consumers that will affect their decision to purchase sea crabs.

## 3. Materials and Methods

### 3.1. Study Area

The selection of Can Tho City, Ho Chi Minh City, and the capital city of Hanoi as the study areas is based on their economic significance, regional diversity, and distinct consumer behaviors regarding seafood consumption. These cities represent Vietnam’s three key regions: the Mekong Delta (Southwest), Southern Vietnam, and Northern Vietnam, providing a comprehensive perspective on consumer purchasing decisions for mud crabs (*Scylla paramamosain*). Can Tho, the economic hub of the Mekong Delta, plays a crucial role in Vietnam’s aquaculture and seafood supply chain. Local consumers have direct access to fresh mud crabs, which influences their price sensitivity and purchasing preferences. Ho Chi Minh City (HCMC), the country’s largest and most economically developed metropolis, has a high demand for seafood across various consumer segments, including restaurants, supermarkets, and online markets. In contrast, Hanoi, the capital of northern Vietnam, relies on mud crab shipments from the south, which may impact availability, pricing, and consumer perceptions compared to those of the other two cities.

### 3.2. Data Collection and Development of Research Scales

#### 3.2.1. Data Collection

This study examined the current state of the research topic in three major cities: Can Tho City, Ho Chi Minh City, and Hanoi Capital, as they represent Vietnam’s key regions. Additionally, Danh and Truc [25] identified these cities as the primary consumers of mud crabs in the Mekong Delta through their application of the chain linkage approach to analyze the mud crab supply chain. The study employed qualitative analysis by synthesizing expert opinions and developing a pilot survey questionnaire, which was refined based on preliminary results before the official survey. A literature review determined that the research model included 21 variables for analysis, requiring a minimum sample size of five samples per variable [68,69], calculated as follows: N × 5 + (N × 5 × 0.1) = 21 × 5 + (21 × 5 × 0.1) = 116. To ensure reliability, the study conducted face-to-face interviews with 200 consumers who purchased mud crabs. The household selection process involved estimating the total number of households on each street (N), calculating the required sample size (n), determining the interval between selected households (k = N/n), and conducting interviews accordingly.

This study examines the current state of mud crab consumption in three major cities representing Vietnam’s key economic regions: Can Tho, Ho Chi Minh City, and Hanoi. Previous research by Danh and Truc [25], using the chain linkage approach, also confirmed that these cities are the primary markets for mud crabs sourced from the Mekong Delta. Methodologically, the study integrates both qualitative and quantitative approaches. The qualitative component involves synthesizing expert opinions and developing a pilot survey questionnaire, which was refined based on preliminary findings. A review of the relevant literature identified 21 key variables to be analyzed. Based on this, the minimum required sample size was calculated as 116 respondents (21 variables × 5 samples per variable for contingency) [70]. To enhance reliability, the study expanded the number of direct interviews to 200 consumers who had purchased mud crabs. A systematic sampling method was employed, involving the following steps: (1) estimating the total number of households along each selected street (N), (2) calculating the required sample size (n), (3) determining the sampling interval as k = N/n, and (4) conducting interviews at intervals based on the calculated k-value [71,72].

#### 3.2.2. Research Scale Development

This study establishes research scales based on the cognitive model, incorporating key factors such as nutritional awareness, food safety awareness, convenience awareness, and price awareness, along with the dependent variable—mud crab purchase decisions. Consumer purchasing decisions are measured using a 5-point Likert scale: (1) Strongly Disagree, (2) Disagree, (3) Neutral, (4) Agree, and (5) Strongly Agree (Likert, 1932). Additionally, the study accounts for demographic factors, including gender (1 = male, 0 = female), age (years), educational attainment (years of schooling), and income (million VND per person per month) [67].

The dependent variable (purchase decision) was measured using a 5-point Likert scale, which is considered an ordinal scale. We employed ordinary least squares (OLS) regression based on the following justifications: First, prior studies in consumer behavior (Carlucci et al., 2015) [42] have demonstrated that OLS can be appropriately applied to Likert-scale data with five or more levels, particularly when the data exhibit characteristics approximating an interval scale. Second, diagnostic tests indicated that the dependent variable displayed adequate dispersion (mean = 3.6, standard deviation = 1.2), and the residuals were approximately normally distributed, satisfying key assumptions for the use of OLS. To further ensure the robustness of our findings, we conducted supplementary analyses using an ordered logit regression model and the Kruskal–Wallis test. Both methods yielded consistent results regarding the statistical significance of the predictors. Nonetheless, we recognize the potential limitations of treating a Likert scale as an interval scale, especially concerning the linearity assumption. To address this issue, we applied robust standard errors and performed sensitivity analyses by collapsing the Likert scale into binary variables.

### 3.3. Data Analysis

#### 3.3.1. Verifying the Reliability Coefficient

The Cronbach’s alpha coefficient is used to identify and exclude inadequate observed variables or scales. The reliability of the scale is assessed using the corrected item–total correlation coefficient, which helps eliminate irrelevant factors from the measurement scale. This coefficient measures the correlation between a specific variable and the average score of other variables within the same scale—a higher coefficient indicates a stronger association with the group. According to Nunnally and Bernstein [73], factors with an item–total correlation coefficient below 0.3 are considered insignificant. To meet reliability requirements, a scale must have a Cronbach’s alpha value greater than 0.6 [69].

The formula for calculating the reliability of the Cronbach’s alpha scale is$$\alpha =\frac{K}{K-1}(1-\frac{\sum_{i=1}^{K}\delta_{Y_{i}}^{2}}{\delta_{X}^{2}})$$
where:*K* is the number of variables (items)—usually the number of questions.Sigma squared is the variance. You can find the statistical terms and formulas using Google, or you may already have a basic understanding of these.*Y* is the component variable.*X* is the total variable.

#### 3.3.2. Exploratory Factor Analysis

According to Trong and Ngoc [69], “Factor analysis is the overarching term for a collection of procedures primarily employed to condense and summarize data.” After scale testing, the remaining observed variables undergo factor analysis to consolidate them into one or more representative factors. The criteria for accepting factor analysis include the following:Eigenvalues (representing the variance explained by each factor) must be greater than 1, with a cumulative variance exceeding 50%.The Kaiser–Meyer–Olkin (KMO) test coefficient assesses the suitability of factor analysis. The KMO value should range between 0.5 and 1; a value below 0.5 indicates that factor analysis may not be appropriate for the data.Factor loading coefficients, which measure the correlation between variables and factors, should be greater than 0.3 to be considered acceptable.

#### 3.3.3. Multiple Regression Analysis

Regression analysis is a statistical technique used to examine the relationship between a dependent variable and multiple independent variables. The regression model is formulated asY_i_ = B_0_ + B_1_ X_1i_ + B_2_ X_2i_ + B_3_ X_3i_ + … + B_P_ X_Pi_ + e_i_

In which:X_pi_ denotes the value of the pi independent variable at the ith observation.B_p_ is the partial regression coefficient.e_i_ is a stochastic independent variable characterized by a normal distribution, with a mean of 0 and constant variance.

Adjusted R^2^ (coefficient of determination): This coefficient measures the proportion of variance in the dependent variable that is explained by the independent variables in the regression model, indicating how well the regression line fits the data. An R^2^ value closer to 1 suggests a strong model fit, while a value near 0 indicates a poor fit. However, R^2^ may provide an overly optimistic assessment, especially when multiple explanatory variables are included. To address this, the adjusted R^2^ is preferred, as it accounts for the number of variables in the model and provides a more accurate measure of model suitability. In this study, the adjusted R^2^ is used to ensure a reliable evaluation of the multivariate linear model, as it mitigates the inflation bias associated with R^2^. Additionally, the F-test in the analysis of variance assesses the overall fit of the model. If the null hypothesis (H_0_) is rejected, it confirms that the multivariate linear regression model is appropriate for the dataset and can be effectively used for analysis.

## 4. Results and Discussion

### 4.1. Attributes of Mud Crab Consumers in Vietnam

A survey of 200 customers across three major cities in Vietnam revealed that 69% of respondents were female (Table 1). The average household size was 3.7 individuals, with the largest household consisting of nine members. The mean monthly income per person was VND 11 million, although income levels varied significantly, ranging from VND 1 million to VND 104 million per month. On average, households allocated approximately VND 11 million per month to food expenses (Table 1). In urban Vietnam, office workers accounted for the majority of consumers (60.4%), and 70% of respondents held a university degree. These findings align with broader economic and demographic trends in Vietnam, where urbanization and rising incomes shape household consumption and spending patterns [74,75]. Moreover, variations in household income and expenditures reflect disparities in economic status and urban–rural differences, as noted in studies on household financial behavior [22,25,43,76].

### 4.2. Domestic Consumer Demand for Fresh Mud Crab in Vietnam

The result showed that among the 200 households that consume fresh mud crabs, the average purchase frequency is 2.1 times per year, with some households buying up to 12 times annually. Of these, 70 households spend an average of VND 1.4 million per year on mud crabs, purchasing approximately 3.5 kg annually. For the 160 households that purchase meat crabs (Y crabs), the average annual consumption is 5.1 kg, with an expenditure of VND 1.6 million. Additionally, 60 households buy crabs with broken claws, purchasing an average of 6 kg per year and spending approximately VND 976,000 annually (Table 2). In terms of preparation methods, consumers most commonly boil fresh crabs (37%), followed by preparing crab hotpot (33%), tamarind crab (16%), salted crab (9%), and crab soup or other dishes (5%) (Figure 2).

Dietary preferences strongly influence fresh crab consumption, with recipes requiring a significant amount of cooked crab being favored for their simplicity, ease of consumption, suitability for all age groups, and ability to retain nutritional value [58,77,78]. Consumers primarily purchase fresh crabs from traditional markets (63.51%), followed by supermarkets (17.57%), farmers (13.51%), warehouses (3.08%), and online platforms (2.03%) (Figure 3). Traditional markets remain the preferred choice for buying fresh products. However, since the outbreak of the COVID-19 pandemic in 2019, consumer purchasing behavior has gradually shifted from traditional markets to modern retail channels, including supermarkets and online shopping [35].

### 4.3. Factors Affecting the Decision to Buy Fresh Mud Crab in Vietnam

#### 4.3.1. Testing the Reliability of Observed Variables

The Cronbach’s alpha reliability coefficient measures the internal consistency and correlation among observed variables, helping to exclude irrelevant examples and reduce extraneous variables in the model (Table 3). Variables with a total correlation coefficient below 0.3 are discarded, while the scale selection criterion requires a Cronbach’s alpha of 0.6 or higher [73]. After assessing the reliability of the observed variables, one independent variable (PR3) and one dependent variable (PD2) were removed (Table 3).

#### 4.3.2. Exploratory Factor Analysis (EFA)—Independent Variable

The results of the exploratory factor analysis (EFA) indicated that out of the original four groups of factors comprising 17 variables (measured on a 5-point Likert scale) influencing the decision to purchase processed crab, one variable (PR3) was removed, leaving 16 variables that met the required criteria (Table 4). Using principal component analysis with varimax rotation, four factor groupings were identified. The analysis demonstrated strong validity, with a Kaiser–Meyer–Olkin (KMO) value of 0.76, exceeding the minimum threshold of 0.5. Additionally, the eigenvalue was 1.069, surpassing the required minimum of 1, and the total variance explained was 70.5%, well above the 50% benchmark (Table 4). The significance level (*p*-value) of 0.000, which is below 0.05, confirms a significant relationship among the observed variables [69].


*Four New Factor Groupings Have Been Established*


Factor 1: Nutritional Awareness—this factor includes four observed variables: NU1 (fresh mud crab ensures freshness), NU2 (fresh mud crab is rich in minerals), NU3 (fresh crab has a naturally sweet taste), and NU4 (fresh mud crab is superior in flavor to canned products). With an average rating of 3.6, this factor significantly influences the decision to purchase fresh crab. Factor 2: Food Safety and Cleanliness Awareness—this factor consists of five observed variables: FHS1 (fresh mud crabs are free from harmful substances), FHS2 (fresh mud crabs have a clear and traceable origin), FHS3 (fresh mud crabs contain no preservatives), and FHS4 (fresh mud crabs are free from parasites) (Table 5). This category has an average rating of 3.8, indicating a moderate influence on the decision to purchase fresh mud crabs. Factor 3: Convenience Perception—this factor includes four observed variables: CO1 (fresh mud crabs offer a variety of choices), CO2 (fresh mud crabs are widely available and easy to purchase), CO3 (fresh mud crabs are convenient to prepare), and CO4 (fresh mud crabs are easy to use after processing). With an average rating of 3.3, this factor has a moderate influence on the decision to purchase fresh mud crabs. Factor 4: Price Perception—this factor consists of three observed variables: PR1 (the price of fresh crab is important to me), PR2 (the price of fresh crab reflects its quality), and PR3 (the price of fresh crab aligns with my household income). This category has an average rating of 3.6, indicating a significant impact on purchasing decisions (Table 5). Overall, the four factors exert varying levels of influence on the decision to purchase fresh crab, with nutritional awareness having the strongest impact. This suggests that consumers prioritize the nutritional value of fresh crab above other considerations, followed by convenience and price. 


*Dependent Variable*


To evaluate the suitability of factor analysis for the original dataset, the Kaiser–Meyer–Olkin (KMO) index and Bartlett’s test were applied. The results indicate a significance level (Sig) of 0.000, leading to the rejection of the null hypothesis (H0). The KMO coefficient of 0.77 falls within the acceptable range of 0.5 to 1 (Table 6), suggesting that the observed variables are sufficiently correlated to justify the use of exploratory factor analysis (EFA). Factor extraction and rotation methods were then performed. The EFA results show that the principal axis factoring extraction method, combined with promax (oblique) rotation, successfully identified a factor comprising three observed variables.

The cumulative variance explained is 0.93, exceeding the minimum threshold of 0.5, while the eigenvalue is 2.8, meeting the requirement of being greater than 1 (Table 7). Additionally, all factor loadings exceed 0.5, confirming that the scale meets the necessary criteria. These findings validate the use of the identified variables in assessing the factors influencing consumers’ decisions to purchase fresh mud crab in Vietnam. For subsequent analyses, the dependent variable will be represented by the mean value of its corresponding observed variables.


*Multiple Regression Analysis*


The residual normality test results indicate that the residuals have a mean of −1 and a standard deviation of 1, which are close to 0. Therefore, the assumption of a normal residual distribution is not violated. Multicollinearity is not a concern, as the multicollinearity test results show that the highest variance inflation factor (VIF) value is 1.05, well below the threshold of 10 (Table 8 and Figure 4). This confirms that the independent variables are not highly correlated with each other, meaning that their interrelationship exerts a minimal impact on the regression model’s explanatory power. A multivariate regression model is employed to analyze the factors influencing the decision to purchase crab. The dependent variable is measured using a Likert scale, while the independent variables are categorized into two groups: cognitive variables and control variables, the latter encompassing personal characteristics and income.

The research hypothesis and model reliability test yielded an adjusted R^2^ of 0.90, indicating that the four independent variables (nutrition awareness, food safety awareness, convenience awareness, and price awareness) and the four control variables (gender, age, food spending, and education level) together explain 90% of the variation in QD (the decision to buy crab). Moreover, all variables are statistically significant at the 5% level, and the F-test result shows a significance level of Sig = 0.000 (<0.05), as indicated by the analysis of variance (Table 9). This confirms that the proposed linear regression model aligns well with the observed data. The independent variables include consumer characteristics (gender, age, income, education level) and four groups of factors from the regression analysis results (Table 9). The values of the four groups of factors represent the factor value scores corresponding to the primary components. The analysis indicates that gender, age, food expenditure, and education level do not influence the decision to select fresh crab products.

Among the component factors, “nutritional awareness” has a significant influence on consumers’ decisions to purchase fresh crab, with a significance level of 1%. The estimated beta coefficient of 0.914 suggests that a one-unit increase in nutritional awareness corresponds to a 91.4% increase in the likelihood of purchasing crab. This finding aligns with previous research on seafood products in general and crab products specifically. The results of this analysis are consistent with those of previous studies [58,61,79]. Descriptive statistics for the “nutritional awareness” group indicate that most crab consumers rate this factor favorably. However, the average score, ranging from 3.6 to 3.64, remains relatively low (Table 8), suggesting that while consumers purchase fresh crab, their understanding of its nutritional value is still limited.

Among the component factors, “awareness of convenience” has a measurable influence on consumers’ decisions to purchase fresh crab. The estimated Beta coefficient is 0.256, with a significance level of 1%, indicating that a one-unit increase in awareness of convenience correlates with a 25.6% increase in the likelihood of purchasing processed crab (Table 9). The analysis results align closely with previous research on factors influencing consumers’ decisions to buy Vietnamese food products [80,81]. Statistical analysis shows that most crab consumers rate the “awareness of convenience” factor positively. However, the average scores, ranging from 3.55 to 3.60, remain relatively low (Table 8), suggesting that while consumers purchase fresh crab, their awareness of its convenience is still limited. Additionally, several other factors such as gender, education level, income, price awareness, and knowledge of food safety and hygiene also influence consumers’ decisions to buy fresh crab. However, their impact lacks statistical significance.

Price perception in Vietnam’s seafood market is shaped by a variety of factors. Culturally, consumers tend to prioritize freshness and product quality over price, particularly when it comes to high-end items such as mud crabs [34]. Market structure also plays a crucial role: 63.5% of seafood transactions occur in traditional markets, where price negotiation is common, leading to lower price sensitivity among consumers [35]. Furthermore, research focusing on consumers with a university education revealed that 70% of them place greater importance on quality than price when purchasing fresh food [37].

Perceptions of food safety in Vietnam’s seafood market are influenced by several interrelated factors. First, information asymmetry plays a critical role. Approximately 86% of mud crabs are purchased live, enabling consumers to visually assess product quality. This direct inspection reduces dependence on abstract or third-party food safety information [43]. Second, limited trust in Vietnam’s seafood certification systems further impacts consumer behavior. Many consumers remain skeptical about the credibility and effectiveness of official safety certifications, casting doubt on their reliability [33]. Finally, cultural heuristics are also at play, particularly in traditional markets where most seafood purchases take place. In these settings, consumers often rely on trust in the vendor as a substitute for formal certification. The seller’s personal reputation and perceived integrity are frequently valued more highly than official safety labels.

Demographic factors influencing crab consumption in Vietnam can be meaningfully understood through several cultural and social dynamics. A primary factor is cultural homogeneity, as crab is a staple in Vietnamese cuisine and enjoys widespread popularity across all demographic groups. This contrasts with patterns observed in Western countries, where demographic characteristics such as age, gender, and income have a stronger influence on seafood consumption behaviors [71]. Furthermore, household roles and responsibilities also shape purchasing behavior. In the study, 69% of respondents were women, reflecting prevailing cultural norms in Vietnam, where women are traditionally responsible for household food purchases, regardless of their income level or age [58].

## 5. Solutions and Recommendations

### 5.1. Basis for Proposing Solutions

This study has developed a new purchase decision scale consisting of four cognitive variables and four control variables. Analysis of the multivariate regression model reveals that two independent variables, namely nutrition awareness and convenience perception, significantly influence the decision to purchase fresh crab. These findings provide a foundation for proposing practical solutions based on the research.

### 5.2. Solutions and Recommendations to Enhance the Value of Vietnam’s Mud Crabs

Based on statistically significant findings related to nutrition perception (β = 0.914) and convenience perception (β = 0.256), this study proposes a set of policy recommendations organized into four main thematic areas. First, there is a clear need to strengthen communication about nutritional value. With 72% of consumers underestimating the nutritional benefits of mud crab, the implementation of standardized nutritional labeling and targeted public awareness campaigns is strongly recommended. Second, the distribution system requires improvement, as only 2.03% of crab purchases are made through online channels. Addressing this issue will require investments in cold chain logistics and the expansion of e-commerce infrastructure to ensure broader and more efficient market access. Third, it is essential to develop a robust quality assurance framework, including the establishment of traceability systems and food safety certification programs tailored to crab farming operations. Fourth, efforts should be made to diversify the market, particularly through the development of value-added, ready-to-eat crab products and by aligning production standards with export market requirements. To guide implementation, a three-phase roadmap is proposed: Short-term (0–2 years)—focus on nutrition labeling reform and public communication strategies. Medium-term (2–5 years)—upgrade supply chain systems and distribution networks. Long-term (5+ years)—develop and expand export markets for mud crab products. These recommendations have been validated for feasibility through comparative policy analysis in similar regional and international contexts.

This study has several limitations that should be considered. Most notably, it focuses solely on urban consumers in three major cities, i.e., Can Tho City, Ho Chi Minh City, and Hanoi Capital; therefore, the findings may not be fully generalizable to rural populations. This limitation arises from several key differences: (1) rural consumers often have direct access to crabs through local aquaculture households; (2) spending on premium seafood products such as mud crabs tends to be significantly lower in rural areas; and (3) there are distinct differences in culinary traditions and food preparation practices between urban and rural regions. To expand the scope and relevance of future research, we propose the following directions: (i) conducting comparative studies between urban and rural areas to examine variations in food safety perceptions, price sensitivity, and preferred purchasing channels; (ii) undertaking surveys across diverse ecological zones (e.g., coastal, delta regions) to evaluate the impact of access to fresh crabs and local consumption habits on consumer behavior; and (iii) implementing in-depth qualitative studies to explore rural consumers’ trust in food quality certifications and the influence of informal distribution networks.

## 6. Conclusions

This study developed a 16-item scale to assess factors influencing consumer purchasing decisions for crabs in the three largest cities of Vietnam. The multiple regression analysis identified two key determinants. First, consumers strongly associate fresh crabs with superior quality, expecting them to provide optimal freshness, high mineral content, excellent flesh texture, and enhanced flavor compared to canned alternatives. To expand the market for fresh crab products, it is crucial to preserve their nutritional value and establish a well-defined nutritional profile for each crab variety at different market stages. Second, convenience plays a significant role in purchasing decisions. Consumers prefer mud crab products that offer a diverse selection, widespread availability, ease of preparation, and simplicity of use after processing.

Based on the regression analysis, two key strategies are recommended for developing new crab products: (1) enhancing nutritional quality to meet consumer expectations for health benefits and (2) strengthening distribution networks and diversifying product offerings to improve accessibility and convenience. To expand the fresh crab market in Vietnam, businesses should focus on maintaining and effectively communicating the high nutritional value of crab products while optimizing distribution and product variety. Implementing these strategies will increase consumer interest and drive market growth, both domestically and internationally.

## Figures and Tables

**Figure 1 foods-14-02198-f001:**
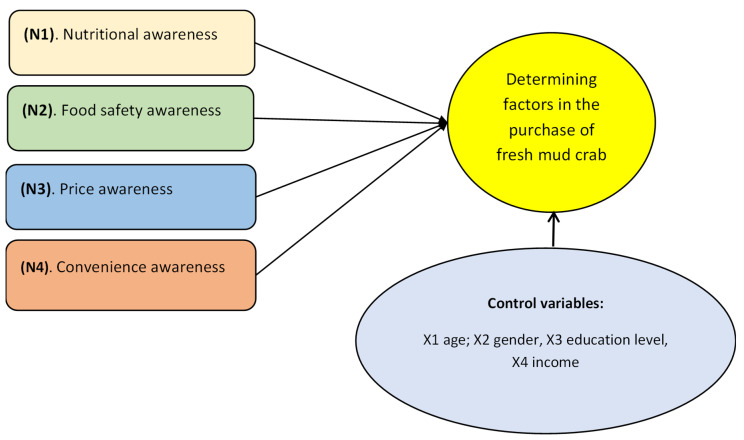
Proposed research model of the key factors influencing the purchase of fresh mud crab.

**Figure 2 foods-14-02198-f002:**
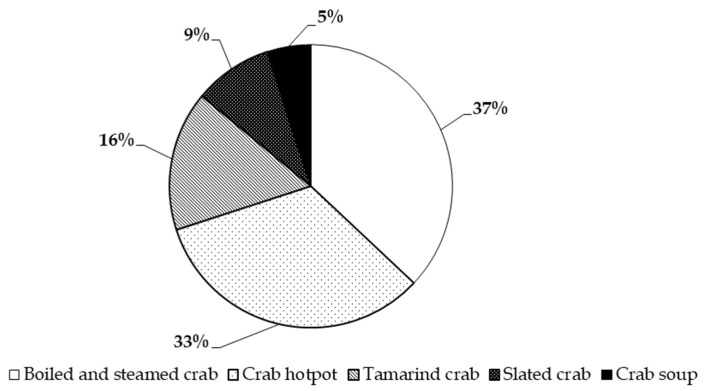
Consumer preferences in three largest cities (Can Tho City, Ho Chi Minh City, and Hanoi Capital) in Vietnam for different fresh mud crab dishes.

**Figure 3 foods-14-02198-f003:**
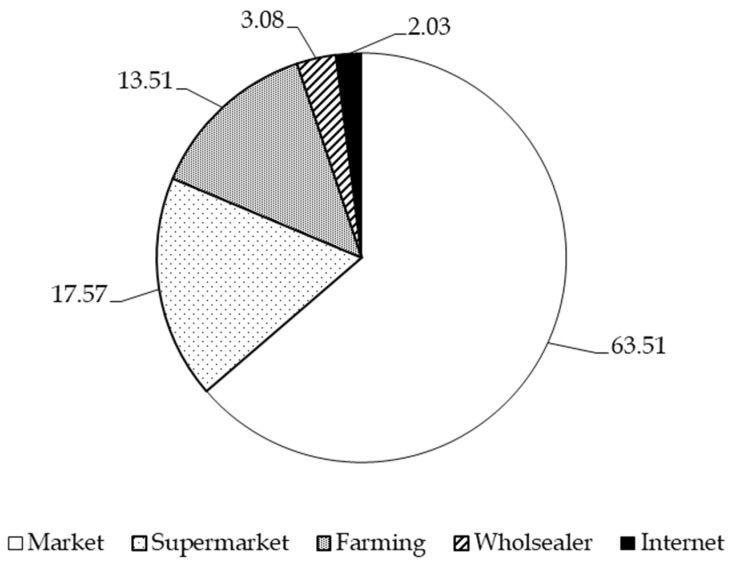
Percentage distribution of consumer preferences for different crab dishes.

**Figure 4 foods-14-02198-f004:**
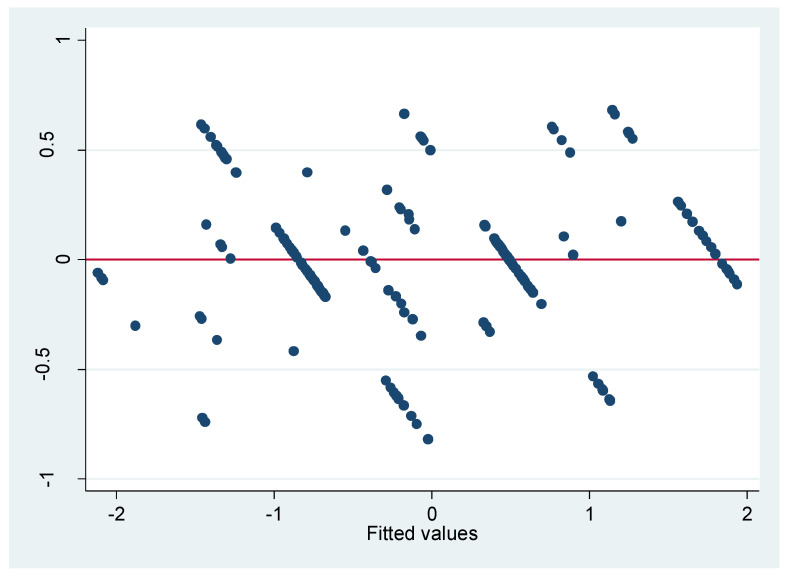
Residual plot.

**Table 1 foods-14-02198-t001:** Characteristics of fresh mud crab consumers in three major cities (Can Tho City, Ho Chi Minh City, and Hanoi Capital) in Vietnam.

Respondent Characteristics	Medium	Standard Deviation	Min	Max
Age (years)	32	8.4	20	62
Number of family members (people)	3.7	1.4	1	9
Female ratio (%)	69			
Male ratio (%)	31			
Education level				
PhD	0.75			
Master’s	7.46			
Bachelor’s	70			
College	8.95			
Intermediate	5.9			
High School	6.94			
Career (%)				
Unemployed	3.7			
Housewife	12			
General laborer	9			
Office staff	60.4			
Teacher	2.9			
Online sales	12			
Income (million VND/person/month)	11	15	1	104
Household food expenses (million VND/month)	11.2	9	2	70

**Table 2 foods-14-02198-t002:** Characteristics of fresh mud crab consumption in three major cities (Can Tho City, Ho Chi Minh City, Hanoi Capital) in Vietnam.

Respondent Characteristics	Medium	Standard Deviation	Min	Max
Number of crab purchases (times/household/year)				
Times	2.1	1.3	1.1	12
Purchase rate of each type of crab (%)				
Crab with roe	53.7			
Crab meat	56.0			
Crab bucket	6.0			
Purchase volume of each type of crab (kg/household/year)				
Crab with roe	3.5	2.8	1.0	20
Crab meat (Y crab)	5.1	2.9	0.5	13
Crab bucket	6.0	2.9	2.0	10
Amount spent on each type of crab (1000 VND/household/year)				
Crab with roe	1454	1040	280	6420
Crab meat (Y crab)	1656	1042	175	5030
Crab bucket	976.0	386.0	420	1560

**Table 3 foods-14-02198-t003:** Cronbach’s alpha test.

Numerical Order	Observation Variable Name	Total Variable Correlation Coefficient	Cronbach’s Alpha Coefficient	Eliminated Variable
NU1	Fresh crab ensures freshness.	0.87	0.92	
NU2	Fresh crab ensures high minerals.	0.86	0.93	
NU3	Fresh crab has sweet meat.	0.87	0.92	
NU4	Fresh crab tastes better than canned products.	0.87	0.92	
FHS1	Fresh crab does not contain toxic substances.	0.74	0.83	
FHS2	Fresh crab ensures clear origin.	0.76	0.82	
FHS3	Fresh crab does not contain preservatives.	0.67	0.84	
FHS4	Fresh crab does not contain parasites.	0.63	0.85	
FHS5	Fresh crab does not cause allergies.	0.66	0.85	
PR1	The price of fresh crab is an important issue for me.	0.88	0.89	
PR2	The price of fresh crab is commensurate with the quality.	0.87	0.82	
PR3	The price of fresh crab is higher than that of shrimp.	0.74	0.84	X
PR4	The price of fresh crab is suitable for my family’s income.	0.87	0.89	
CO1	Fresh crab has a variety of types to choose from.	0.81	0.94	
CO2	Fresh crab is easy to find everywhere.	0.82	0.93	
CO3	Fresh crab is convenient for processing.	0.83	0.96	
CO4	Fresh crab is easy to use after processing.	0.83	0.90	
PD1	I decided to buy crab because of its high nutrition.	0.91	0.91	
PD2	I decided to buy crab because of food safety.	0.77	0.96	X
PD3	I decided to buy crab sea because the price is suitable for family.	0.90	0.90	
PD4	I decided to buy sea crab because of the high convenience.	0.91	0.90	

Note: X refers to the excluded variables, which were not significant.

**Table 4 foods-14-02198-t004:** Matrix of factors influencing an individual’s decision to consume fresh crab in the largest cities (Can Tho City, Ho Chi Minh City, and Hanoi Capital) in Vietnam.

Observation Variable	Factors			
	1	2	3	4
Fresh mud crab ensures freshness.	0.90			
2. Fresh mud crab ensures high minerals.	0.89			
3. Fresh mud crab has sweet meat.	0.91			
4. Fresh mud crab tastes better than canned products.	0.89			
5. Fresh mud crab does not contain toxic substances.		0.84		
6. Fresh mud crab ensures clear origin.		0.85		
7. Fresh mud crab does not contain preservatives.		0.79		
8. Fresh mud crab does not contain parasites.		0.77		
9. Fresh mud crab does not cause allergies.		0.78		
10. Fresh mud crab has a variety of types to choose from.			0.87	
11. Fresh mud crab is easy to find everywhere.			0.86	
12. Fresh mud crab is convenient for processing.			0.89	
13. Fresh mud crab is easy to use after processing.			0.88	
14. The price of fresh mud crab is an important issue for me.				0.94
15. The price of fresh mud crab is commensurate with the quality.				0.96
16. The price of fresh mud crab is suitable for my family’s income.				0.93
KMO coefficients	0.706			

**Table 5 foods-14-02198-t005:** Groups of factors influencing people’s decision to purchase fresh crab in three largest cities (Can Tho City, Ho Chi Minh City, and Hanoi Capital), Vietnam.

Group of Factors	Factor Name	Medium	Evaluate **
N1	Nutritional awareness	3.6	affect
N2	Food safety awareness	3.8	affect
N3	Perception of convenience	3.6	affect
N4	Price awareness	3.3	neutral

Note: ** Evaluation based on 5-point Likert scale: 1–1.8: strongly unaffecting/strongly disagree; 1.8–2.6: no affect/disagree; 2.6–3.4: neutral; 3.4–4.2: affect/agree; and 4.2–5: strongly affecting/strongly agree.

**Table 6 foods-14-02198-t006:** KMO and Bartlett’s test tourist satisfaction scale results for the three largest cities (Can Tho City, Ho Chi Minh City, and Hanoi Capital) in Vietnam.

KMO Coefficient		0.77
Bartlett’s test	Chi-square value	732
	df	3.0
	Sig—observational significance level	0.000

**Table 7 foods-14-02198-t007:** Factor analysis results for the tourist satisfaction scale in the three largest cities (Can Tho City, Ho Chi Minh City, and Hanoi Capital) in Vietnam.

	Load Factor
Purchase decision 1	0.97
Purchase decision 3	0.97
Purchase decision 4	0.95
Eigenvalue	2.80
Cumulative variance	0.93

**Table 8 foods-14-02198-t008:** VIF multicollinearity test.

Variable Name	VIF	1/VIF
Age (years)	1.14	0.87
Spending	1.1	0.91
Gender	1.03	0.97
Education	1.03	0.97
N1 Nutritional awareness	1.01	0.99
N2 Food safety awareness	1.04	0.96
N3 Perception of convenience	1.00	0.96
N4 Price awareness	1.03	0.99
Mean VIF	1.05	

**Table 9 foods-14-02198-t009:** Factors influencing consumers’ intention to purchase processed crab in the three largest cities (Can Tho City, Ho Chi Minh City, and Hanoi Capital) in Vietnam.

Symbol	Variable Name	Estimated Coefficient β	Standard Error	*p*-Value
X1	Age (years)	0.001 ^ns^	0.003	0.659
X2	Gender (1—male; 0—female)	−0.02 ^ns^	0.049	0.650
X3	Education (years of schooling)	−0.08 ^ns^	0.014	0.589
X4	Income (million VND/month)	−0.04 ^ns^	0.031	0.235
N1	Nutritional awareness	0.914 ***	0.022	0.000
N2	Food safety awareness	0.036 ^ns^	0.023	0.119
N3	Perception of convenience	0.256 ***	0.022	0.000
N4	Price awareness	−0.014 ^ns^	0.022	0.547
	Constant	0.193 ^ns^	0.247	0.435
Y	Decision to buy fresh crab			
Number of observations			200	
*p*-value			0.000	
Coefficient of determination R^2^ (%)			90	

Note: n = 200: number of interviewed mud crab consumers; (*** and ns are at the 1%, 5%, 10% and not significant levels, respectively), ns: non-significant.

## Data Availability

The original contributions presented in this study are included in the article. Further inquiries can be directed to the corresponding author.

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
