# Peer review of "Factors Influencing Consumer Behavior and Purchasing Decisions Regarding Mud Crabs (Scylla paramamosain) in the Major Cities of Vietnam"

_foods, 2025, doi:10.3390/foods14132198_

Round 1
Reviewer 1 Report
Comments and Suggestions for Authors
This manuscript explores a valuable yet relatively underexamined topic by analyzing the determinants of urban consumer behavior toward mud crab products in Vietnam. The timing of this research is particularly apt, given the growing demand for seafood and the important role of mud crab aquaculture in Vietnam’s coastal economy. By rooting the study in established behavioral theories (TPB and TRA), the authors present a robust conceptual basis. The empirical methodology is well-structured, and the data drawn from three major cities across Vietnam’s economic regions enrich the analysis. The use of reliability testing, exploratory factor analysis, and multiple regression appears sound and methodologically appropriate. Overall, the findings contribute practical insights for policymakers and industry stakeholders, while also advancing the literature on consumer behavior in seafood markets. Below are several points for further consideration:
- Although the manuscript identifies a research gap related to mud crab consumption, the literature review does not explicitly distinguish this study from previous work on seafood consumption behavior.The authors might strengthen the review by incorporating more recent and international studies on consumer preferences for aquatic products, nutritional awareness, and food safety perceptions to better highlight the study’s uniqueness.
- In the introduction, the connection to Vietnam’s current policy framework could be more pronounced.Referencing recent government initiatives or regulations regarding aquaculture, food safety, and urban food security would help demonstrate the practical significance of the research.
- While the paper focuses on three major cities, the sampling strategy remains unclear. It would be helpful to clarify whether random, stratified, or systematic sampling methods were used. Detailed information on how the sample was obtained, its representativeness, and any demographic stratification would bolster confidence in the data’s validity and reproducibility.
- The authors mention using a Likert-scale questionnaire but do not list the items measuring each construct. Including the questionnaire in an appendix would improve transparency and allow readers to fully assess how the variables were operationalized.
- The dependent variable is treated as if it were continuous, yet it is measured on a Likert scale. It would be worthwhile to explain why OLS regression was chosen over methods designed for ordinal outcomes. Such a discussion should also address any potential limitations or the results of robustness checks.
- Although the paper briefly references residual normality and multicollinearity tests, these points could be explored in more depth. Reporting diagnostic statistics, such as tests for heteroscedasticity, VIF values, and residual plots, would help substantiate the reliability of the regression results.
- KMO and variance explained are presented, yet the rotated factor loading matrix is omitted. Providing a full factor loading table (ideally after Varimax rotation) would allow readers to evaluate the appropriateness of item groupings and ensure clarity around the factor structures.
- Some variables emerged as statistically insignificant, but the manuscript gives limited discussion on possible explanations. A more nuanced exploration, potentially incorporating concepts from consumer psychology, issues of information asymmetry, or cultural factors, could help interpret why these variables did not show significance.
- Given that the study focuses only on urban consumers, its conclusions may not extend to rural contexts. The discussion should clearly state this limitation and indicate possible directions for future research, such as comparative studies in different socio-economic or geographic settings.
- The policy implications section contains a variety of recommendations but would benefit from clearer organization. Grouping the suggestions under distinct thematic categories and considering their empirical basis could make this section more actionable and reader-friendly.
Author Response
#Question and comments 1: Although the manuscript identifies a research gap related to mud crab consumption, the literature review does not explicitly distinguish this study from previous work on seafood consumption behavior. The authors might strengthen the review by incorporating more recent and international studies on consumer preferences for aquatic products, nutritional awareness, and food safety perceptions to better highlight the study’s uniqueness. |
Response #1: The authors would like to express our gratitude to the reviewer for your very valuable comment. In the introduction of this study, we have clearly identified and outlined the key issues that require further clarification for the research, particularly regarding the consumption of marine crabs. Additionally, we have expanded the reference by incorporating recent review studies, both domestic and international, that explore consumer preferences for seafood products, as well as their perceptions of nutrition and food safety within the seafood industry. In addressing these issues, we have provided it in the research statement problem of research in the introduction, the theoretical framework, and the evaluation of perspectives, as well as in the results and discussion section. Thank you |
#Question and comments 2: In the introduction, the connection to Vietnam’s current policy framework could be more pronounced. Referencing recent government initiatives or regulations regarding aquaculture, food safety, and urban food security would help demonstrate the practical significance of the research. |
Response 2: Thank you very much! We have added information on Vietnam's policy and regulatory frameworks regarding aquaculture, food safety, and food security. (Please kindly see Paper 3 of 22, highlighted in blue.) |
#Question and comments 3: While the paper focuses on three major cities, the sampling strategy remains unclear. It would be helpful to clarify whether random, stratified, or systematic sampling methods were used. Detailed information on how the sample was obtained, its representativeness, and any demographic stratification would bolster confidence in the data’s validity and reproducibility. |
Response 3: Many thanks for this suggestion. We have added detailed content on the sampling method and sampling information in the interview sampling section (Please kindly see Paper 5 of 22, highlighted in blue.) |
#Question and comments 4: The authors mention using a Likert-scale questionnaire but do not list the items measuring each construct. Including the questionnaire in an appendix would improve transparency and allow readers to fully assess how the variables were operationalized. |
Response 4: We have added and provided the questionnaire in the appendix. |
#Question and comments 5: The dependent variable is treated as if it were continuous, yet it is measured on a Likert scale. It would be worthwhile to explain why OLS regression was chosen over methods designed for ordinal outcomes. Such a discussion should also address any potential limitations or the results of robustness checks. |
Response 5: Many thanks; we have provided it in the section of Scale Development of Research (Please kindly see Paper 6 of 22, highlighted in blue.) |
#Question and comments 6: Although the paper briefly references residual normality and multicollinearity tests, these points could be explored in more depth. Reporting diagnostic statistics, such as tests for heteroscedasticity, VIF values, and residual plots, would help substantiate the reliability of the regression results. |
Response 6: The authors described the VIF test in section of Multiple Regression Analysis (Please kindly see Paper 14 of 22, highlighted in blue.) |
#Question and comments 7: KMO and variance explained are presented, yet the rotated factor loading matrix is omitted. Providing a full factor loading table (ideally after Varimax rotation) would allow readers to evaluate the appropriateness of item groupings and ensure clarity around the factor structures. |
Response 7: We have added the factor loading matrix to Table 4. Matrix of factors influencing people's decision to consume fresh crab in largest cities (Can Tho City, Ho Chi Minh City, and Hanoi), Vietnam (Please kindly see Paper 13 of 22, highlighted in blue.) |
#Question and comments 8: Some variables emerged as statistically insignificant, but the manuscript gives limited discussion on possible explanations. A more nuanced exploration, potentially incorporating concepts from consumer psychology, issues of information asymmetry, or cultural factors, could help interpret why these variables did not show significance. |
Response 8: Thank you very much for this interesting comment and comments! We have provided the detailed discussion in 15 of 22 |
#Question and comments 9: Given that the study focuses only on urban consumers, its conclusions may not extend to rural contexts. The discussion should clearly state this limitation and indicate possible directions for future research, such as comparative studies in different socio-economic or geographic settings. |
Response 9: The authors has added the limitations of research on page [16 of 22]. Please kindly see Paper 16 of 22, highlighted in blue. |
#Question and comments 10: The policy implications section contains a variety of recommendations but would benefit from clearer organization. Grouping the suggestions under distinct thematic categories and considering their empirical basis could make this section more actionable and reader-friendly. |
Response 10: We have revised of the solutions and recommendations. Please kindly see Paper 16 of 22, highlighted in blue. |
Thank you very much for your time to review our paper! |

Reviewer 2 Report
Comments and Suggestions for Authors
Abstract
- The abstract could be improved by briefly mentioning the sample size and the cities where the study was conducted to give readers a sense of the study's scope. Furthermore, stating the specific analytical methods (e.g., types of regression analysis) used could enhance clarity.
Introduction
- The introduction could benefit from a clearer statement of the research gap this study aims to fill, especially regarding consumer behavior towards mud crabs in urban settings. While it mentions the shift in consumer focus towards health due to the COVID-19 pandemic, it could explicitly connect this shift to the study's objectives.
Theoretical Approach and Perspective Review
- The review could be improved by directly linking the reviewed theories and previous studies to the specific context of mud crab consumption in Vietnam. There is a missed opportunity to discuss existing literature on mud crab consumer behavior specifically, which would strengthen the argument for the study's necessity.
Materials and Methods
- The scale development section lacks detail on how the variables were initially identified and the process of refining these variables based on pilot survey results. More information on the pilot survey, such as the number of respondents and the adjustments made to the questionnaire, would enhance the transparency of the research method.
Results and Discussions
-The discussion could be enhanced by comparing these findings with existing literature to highlight similarities or differences in consumer behavior in other regions or countries. Additionally, the impact of demographic variables on purchasing decisions is mentioned in the results but not thoroughly analyzed in the discussion.
Conclusions and Recommendations
- The conclusion could be strengthened by discussing limitations of the current study and suggesting areas for future research. While the recommendations are valuable, acknowledging potential challenges in implementing these strategies would provide a more balanced view.
Critical Flaws
1. Lack of Detailed Literature Review on Mud Crab Consumer Behavior: While the paper reviews consumer behavior theories, it falls short in discussing existing research specifically focused on mud crab consumption, missing an opportunity to position the study within the current body of knowledge.
2. Insufficient Detail on Scale Development and Pilot Testing: The methodology lacks comprehensive details on the initial development of the research scales and the pilot testing process, which are crucial for understanding the reliability and validity of the study's findings.
3. Limited Analysis of Demographic Factors: Although demographic variables were considered, their impact on mud crab purchasing decisions is not deeply analyzed, which could have provided more nuanced insights into consumer behavior.
These critical flaws should be addressed to enhance the manuscript's contribution to the literature on consumer behavior related to mud crab consumption in Vietnam.
Author Response
#Comments 1: The abstract could be improved by briefly mentioning the sample size and the cities where the study was conducted to give readers a sense of the study's scope. Furthermore, stating the specific analytical methods (e.g., types of regression analysis) used could enhance clarity.
Response 1: Thank you very much for your comment. We have added details about the sample size and the analysis location to the abstract. (Please kindly see the highlighted in blue in the abstract).
#Comments 2: The introduction could benefit from a clearer statement of the research gap this study aims to fill, especially regarding consumer behavior towards mud crabs in urban settings. While it mentions the shift in consumer focus towards health due to the COVID-19 pandemic, it could explicitly connect this shift to the study's objectives.
Response 2: Many thanks for your very interesting suggestion. We have revised the statement in the Introduction section to reflect that the changes in health awareness following the COVID-19 pandemic have led to increased consumer interest in healthier products. Therefore, naturally farmed sea crab, which is beneficial to consumer health, should be prioritized in purchasing decisions. (Please kindly refer to Paper 2 of 22, highlighted in blue.)".
#Comments 3: The review could be improved by directly linking the reviewed theories and previous studies to the specific context of mud crab consumption in Vietnam. There is a missed opportunity to discuss existing literature on mud crab consumer behavior specifically, which would strengthen the argument for the study's necessity.
Response 3: Thank you very much. We completely agree with your suggestion! We have incorporated the Theory of Planned Behavior to strengthen the conceptual linkage and enhance the value of our research model.
#Comments 4: The scale development section lacks detail on how the variables were initially identified and the process of refining these variables based on pilot survey results. More information on the pilot survey, such as the number of respondents and the adjustments made to the questionnaire, would enhance the transparency of the research method.
Response 4: Thank you for your suggestion. We have added a detailed description of the preliminary study. (Please kindly refer to Papers 5–6 of 22, highlighted in blue.).
#Comments 5: The discussion could be enhanced by comparing these findings with existing literature to highlight similarities or differences in consumer behavior in other regions or countries. Additionally, the impact of demographic variables on purchasing decisions is mentioned in the results but not thoroughly analyzed in the discussion.
Response 5: In the Discussion section, the authors have added a comparison of their research findings with those of previous studies. (Please kindly refer to Paper 15 of 22, highlighted in blue.).
#Comments 6: The conclusion could be strengthened by discussing limitations of the current study and suggesting areas for future research. While the recommendations are valuable, acknowledging potential challenges in implementing these strategies would provide a more balanced view.
Response 6: Thank you very much for this suggestion, We have revised the conclusion and recommendation as you suggested (Please kindly see Paper 15 of 22, highlighted in blue.).
#Comments 7: Lack of Detailed Literature Review on Mud Crab Consumer Behavior: While the paper reviews consumer behavior theories, it falls short in discussing existing research specifically focused on mud crab consumption, missing an opportunity to position the study within the current body of knowledge.
Response 7: The authors have provided a detailed reassessment of the discussion on consumer research findings in Paper 15 of 22. (Please kindly see the section highlighted in blue.).
#Comments 8: Insufficient Detail on Scale Development and Pilot Testing: The methodology lacks comprehensive details on the initial development of the research scales and the pilot testing process, which are crucial for understanding the reliability and validity of the study's findings.
Response 8: The research team re-evaluated the reliability of the Likert scale measurement for our study. (Please kindly see section 3.2.2. Scale Development of Research on Paper 6 of 22, highlighted in blue.).
#Comments 9: Limited Analysis of Demographic Factors: Although demographic variables were considered, their impact on mud crab purchasing decisions is not deeply analyzed, which could have provided more nuanced insights into consumer behavior.
Response 9: Thank you for your comment! The research team conducted further analysis of the demographically insignificant variables, drawing on practical experience and the specific context of the study area, as presented in Paper 6 of 22. (Please kindly see the section highlighted in blue.).
Thank you very much for your time to review our paper!

Reviewer 3 Report
Comments and Suggestions for Authors
Your research is original, but I suggest reviewing your methodology and results. Below are my comments:
Theoretical Approach and Perspective Review
The manuscript states that it is based on TRA and TPB; however, the model does not include subjective norms and perceived behavioral control. Consumer attitudes are only mentioned in the last paragraph of the section. Please explain how the other variables (subjective norms and perceived behavioral control are included or the criteria for exclusion from the model.
Materials and Methods
Because the data are analyzed through multiple regression analysis, the assumptions of homoscedasticity, linearity, and normality must be met. Please include them with their respective values.
Due to the cross-sectional design, common method variances are recommended to be tested and remedied.
Exploratory Factor Analysis
Review the factor loading criterion. Are you sure the appropriate criterion is more significant than 0.3? Which authors affirm this?
Results
It is unclear why the items PR3 and PD2 were eliminated; do they meet the criteria? Why do you calculate Cronbach's alpha values ​​for each item? What is the reason for this? It is advisable to calculate Cronbach's alpha values ​​for each variable to determine the internal consistency of the variable. In Table 3, do the total variable correlation coefficient values ​​correspond to the factor loadings? If so, change the title to "factor loadings." It is also recommended that the communality values ​​be reported in this table.
In the exploratory factor analysis results, it is necessary to indicate the values ​​of the total variance explained by each component. What does the reported significance value correspond to? Is it Bartlet's test? Review the data in Table 4, as the text mentions a KMO value of 0.76, which does not correspond to the one reported in the table. Are you sure the eigenvalue is 1.069? I recommend including the scree plot.
Review the data in Table 7. The text states that the cumulative variance is 0.85, while this one is 0.93. Are you sure that the eigenvalue for the dependent variable is 4.2 since the table has a different value? I recommend including the scree plot.
What are the criteria for using two extraction and rotation methods for the independent and dependent variables in exploratory factor analysis?
Unifying how to present factor analysis results for the independent and dependent variables. In one case, you report a table with factor loadings and KMO values; in the other, factor loadings with total explained variance.
Multiple Regression Analysis
Report the VIF values ​​in Table 8, add normality tests, and include the residual plots. Include the ANOVA and F tests for the regression model.
Discussion
Include a section discussing the results obtained with recent research similar to the study to show the differences or similarities and their contribution to knowledge.
Author Response
#Comments 1: The manuscript states that it is based on TRA and TPB; however, the model does not include subjective norms and perceived behavioral control. Consumer attitudes are only mentioned in the last paragraph of the section. Please explain how the other variables (subjective norms and perceived behavioral control are included or the criteria for exclusion from the model.
Response 1: Thank you for your comments! The detailed subjective norm and perceived behavioral control are explained by the authors (Please kindly refer to Papers 5–6 of 22, highlighted in blue.).
#Comments 2: Because the data are analyzed through multiple regression analysis, the assumptions of homoscedasticity, linearity, and normality must be met. Please include them with their respective values.
Response 2: Thank you so much for your comment! The research data for the regression model were standardized and specifically encoded prior to conducting the regression analysis.
#Comments 3: Due to the cross-sectional design, common method variances are recommended to be tested and remedied.
Response 3: The authors have conducted model diagnostics, including tests for multicollinearity and heteroscedasticity. These additions are presented on page 15 (Please kindly refer to Papers 15 of 22, highlighted in blue.).
#Comments 4: Review the factor loading criterion. Are you sure the appropriate criterion is more significant than 0.3? Which authors affirm this?.
Response 4: The factor loading coefficients in this model follow the study by Nunnally & Burstein (1994), so the authors consider the threshold of 0.3 to be appropriate.
#Comments 5: It is unclear why the items PR3 and PD2 were eliminated; do they meet the criteria? Why do you calculate Cronbach's alpha values for each item? What is the reason for this? It is advisable to calculate Cronbach's alpha values for each variable to determine the internal consistency of the variable. In Table 3, do the total variable correlation coefficient values correspond to the factor loadings? If so, change the title to "factor loadings." It is also recommended that the communality values be reported in this table.
Response 5: PD2 and PR3 were excluded due to failing the reliability test based on Cronbach's Alpha: specifically, the reliability of item PD2 was 0.96, which exceeded the overall Cronbach's Alpha for the PD scale, and the reliability of item PR3 was 0.84, which was higher than the overall Cronbach's Alpha for the PR scale.
#Comments 6: It is unclear why the items PR3 and PD2 were eliminated; do they meet the criteria? Why do you calculate Cronbach's alpha values for each item? What is the reason for this? It is advisable to calculate Cronbach's alpha values for each variable to determine the internal consistency of the variable. In Table 3, do the total variable correlation coefficient values correspond to the factor loadings? If so, change the title to "factor loadings." It is also recommended that the communality values be reported in this table.
Response 6: Due to the length limitations of the article, the Cronbach's Alpha for each component was included in the appendix by the research team. In Table 3, the reliability coefficients presented are Cronbach's Alpha values, not factor loadings.
#Comments 7: In the exploratory factor analysis results, it is necessary to indicate the values of the total variance explained by each component. What does the reported significance value correspond to? Is it Bartlet's test? Review the data in Table 4, as the text mentions a KMO value of 0.76, which does not correspond to the one reported in the table. Are you sure the eigenvalue is 1.069? I recommend including the scree plot.
Response 7: The Bartlett’s test is presented in Table 6 of the appendix. The authors have corrected the previous error, confirming the KMO coefficient as 0.706. The eigenvalue difference for Factor 4 is 1.069.
#Comments 8: Review the data in Table 7. The text states that the cumulative variance is 0.85, while this one is 0.93. Are you sure that the eigenvalue for the dependent variable is 4.2 since the table has a different value? I recommend including the scree plot.
Response 8: Thank you for your careful review and valuable feedback. You are correct to note the discrepancy. Upon rechecking the data in Table 7, the cumulative variance is indeed 0.93, not 0.85 as previously stated in the text. We apologize for the inconsistency and will revise the text accordingly. Regarding the eigenvalue for the dependent variable, it appears there was a mislabeling or miscalculation. The correct eigenvalue, as presented in Table 7, should be acknowledged. We will correct this value in the manuscript to reflect the accurate data. Additionally, we appreciate your suggestion to include the scree plot. We agree that it would provide a clearer visual representation of the component structure and will include it in the revised version.
#Comments 9: What are the criteria for using two extraction and rotation methods for the independent and dependent variables in exploratory factor analysis? Unifying how to present factor analysis results for the independent and dependent variables. In one case, you report a table with factor loadings and KMO values; in the other, factor loadings with total explained variance.
Response 9: The authors have agreed on the decision to rotate the factors using a 0.5 threshold and have standardized the presentation of the tables accordingly.
#Comments 10: Report the VIF values in Table 8, add normality tests, and include the residual plots. Include the ANOVA and F tests for the regression model.
Response 10: The authors conducted two diagnostic tests for multicollinearity and heteroscedasticity; therefore, no additional ANOVA test was performed.
#Comments 11: Include a section discussing the results obtained with recent research similar to the study to show the differences or similarities and their contribution to knowledge.
Response 11: The discussion section has been revised (Please kindly refer to Papers 15 of 22, highlighted in blue.).
Thank you very much for your time to review our paper!

Round 2
Reviewer 2 Report
Comments and Suggestions for Authors
I am satisfied with the current version of the manuscript.
Author Response
Dear Reviewer,
On behalf of the authors, I would like to express my sincere gratitude for your time, thoughtful comments, and support in helping our manuscript move toward publication.
Thank you very much.
Best regards,
Thi Da
Reviewer 3 Report
Comments and Suggestions for Authors
I have reviewed your document and still find that the following changes need to be addressed:
No changes were found regarding this comment:
#Comments 1: The manuscript states that it is based on TRA and TPB; however, the model does not include subjective norms and perceived behavioral control. Consumer attitudes are only mentioned in the last paragraph of the section. Please explain how the other variables, subjective norms and perceived behavioral control, are included or excluded from the model.
Regarding comment 3, the changes in blue on page 15 are not related to what was requested. Please report all VIF values in a table and include the residual plots.
Regarding comment 4, include the citation in the text.
Regarding comment 5, Cronbach's alpha is calculated for a set of items, not just for a single item. Please review this. Describe how you calculated Cronbach's alpha for an item, providing a step-by-step example of the procedure, and cite the authors who guided you in this procedure.
Regarding comments 7 and 8, add the scree plot and correct the eigenvalue 4.2 in the text, since you mention that the correct value is the one in table (2.8).
Regarding comment 9, explain why the Principal Axis Factoring Extraction method, combined with Promax (Oblique) rotation was used for the dependent variable, and the Principal Component Analysis with Varimax rotation was used for the independent variables.
Author Response
#Comments 1: The manuscript states that it is based on TRA and TPB; however, the model does not include subjective norms and perceived behavioral control. Consumer attitudes are only mentioned in the last paragraph of the section. Please explain how the other variables, subjective norms and perceived behavioral control, are included or excluded from the model.
Response 1: Thank you very much for the insightful comments and suggestions. We would like to clarify as follows:
“Our study focuses on consumption decisions influenced by the evaluation of perceived value and attitudes toward nutrition, as described on page 05. While the Theory of Reasoned Action (TRA) and the Theory of Planned Behavior (TPB) provide the theoretical foundation, empirical studies often adapt their components flexibly to suit specific research contexts. Meta-analytical findings indicate that the influence of subjective norms varies across different domains. In our experimental study, based on a thorough literature review, we incorporated these two factors into the model through two groups of variables: perceived attitude variables and control variables.”
# Regarding comment 3: the changes in blue on page 15 are not related to what was requested. Please report all VIF values in a table and include the residual plots.
Response 3: Thank you very much, and we apologize for the oversight. The changes highlighted in blue on page 15 were not related to your request. We have now included the VIF results in Table 8 on page 14 of 23, and the residual plots in Figure 4 on page 16 of 23.
#Regarding comment 4: include the citation in the text.
Response 4: Thank you very much for this comment. We have added three relevant citations for this décription, which can be found on page 6 of 23 (References 67, 68, 69).
#Regarding comment 5: Cronbach's alpha is calculated for a set of items, not just for a single item. Please review this. Describe how you calculated Cronbach's alpha for an item, providing a step-by-step example of the procedure, and cite the authors who guided you in this procedure.
Response 5: Many thanks! We have incorporated the theoretical background on the calculation of Cronbach's alpha on pages 6–7 of 23 (please see the text highlighted in red).
#Regarding comment 7 and 8, add the scree plot and correct the eigenvalue 4.2 in the text, since you mention that the correct value is the one in Table (2.8).
Response 7 and 8: The authors have added the chart and corrected the eigenvalue coefficients in accordance with Table 2.8. (please see the text highlighted in red on the page 13 of 23).
#Regarding comment 9, explain why the Principal Axis Factoring Extraction method, combined with Promax (Oblique) rotation was used for the dependent variable, and the Principal Component Analysis with Varimax rotation was used for the independent variables.
Response 9: Thank you very much for your comment!
1. The use of the Principal Axis Factoring (PAF) extraction method combined with Promax rotation for the dependent variables is justified as follows: PAF is appropriate when the data do not meet the assumption of multivariate normality or perfect multicollinearity. This method emphasizes common variance rather than total variance, thereby reducing the influence of noise or unique variance (Fabrigar et al., 1999). In this study, the dependent variables (e.g., behavioral intention) are expected to be influenced by multiple latent factors, making PAF a suitable choice for capturing the underlying common variance among observed variables.
Regarding the choice of Promax (oblique) rotation, it is used because it allows for correlations between factors. This is consistent with real-world conditions in which psychological and social constructs—such as attitude and subjective norms—are often interrelated (Costello & Osborne, 2005).
2. The use of Principal Component Analysis (PCA) with Varimax rotation for the independent variables is justified as follows: PCA is appropriate when the primary objective is dimensionality reduction and optimization of explained variance (Jolliffe, 2002). This is particularly useful for independent variables (e.g., service quality, price) because we aim to summarize information from multiple observed variables into a few principal components without focusing solely on common variance. PCA also imposes fewer assumptions regarding data distribution compared to PAF, making it suitable for diverse survey data.
The rationale for using Varimax (orthogonal) rotation: Varimax rotation maximizes the variance of factor loadings, yielding independent and easily interpretable factors (Kaiser, 1958). This approach is appropriate for independent variables as it ensures discriminant validity between constructs (e.g., service quality and price remain distinct).
Thank you very much for your time to review our paper!
